# Bidirectional Optical Neural Networks Based on Free-Space Optics Using Lens Arrays and Spatial Light Modulator

**DOI:** 10.3390/mi15060701

**Published:** 2024-05-25

**Authors:** Young-Gu Ju

**Affiliations:** Department of Physics Education, Kyungpook National University, 80 Daehak-ro, Buk-gu, Daegu 41566, Republic of Korea; ygju@knu.ac.kr; Tel.: +82-(53)-9505894

**Keywords:** optical neural network, optical computer, backpropagation, neural network, lens array, free-space optics

## Abstract

This paper introduces a novel architecture—bidirectional optical neural network (BONN)—for providing backward connections alongside forward connections in artificial neural networks (ANNs). BONN incorporates laser diodes and photodiodes and exploits the properties of Köhler illumination to establish optical channels for backward directions. Thus, it has bidirectional functionality that is crucial for algorithms such as the backpropagation algorithm. BONN has a scaling limit of 96 × 96 for input and output arrays, and a throughput of 8.5 × 10^15^ MAC/s. While BONN’s throughput may rise with additional layers for continuous input, limitations emerge in the backpropagation algorithm, as its throughput does not scale with layer count. The successful BONN-based implementation of the backpropagation algorithm requires the development of a fast spatial light modulator to accommodate frequent data flow changes. A two-mirror-like BONN and its cascaded extension are alternatives for multilayer emulation, and they help save hardware space and increase the parallel throughput for inference. An investigation into the application of the clustering technique to BONN revealed its potential to help overcome scaling limits and to provide full interconnections for backward directions between doubled input and output ports. BONN’s bidirectional nature holds promise for enhancing supervised learning in ANNs and increasing hardware compactness.

## 1. Introduction

In recent decades, significant effort has been invested in developing optical computers for real-time data processing, owing to their potential advantages in terms of speed and parallelism over electronic computers [1,2,3]. However, optical computing has not surpassed digital computing due to the faster progress and greater power, ease of use, and flexibility of its competitor [4]. While digital optics lacks proper components to compete, combining optics and electronics could be fruitful, with optics being used for tasks where it excels. Free-space optics [3] and smart pixels [5] are promising options for overcoming the limitations of electronic processors, with the former being especially beneficial for artificial neural networks [6]. Purely electrical neural networks have a complex topology of connections as the number of input and output nodes increases. Moreover, the complex topology and proximity between electrical wires may cause significant electromagnetic crosstalk or noise, especially at high clock rates [3,4]. Unlike electrical interconnections, light paths in free-space optics can cross freely without electromagnetic crosstalk noise between channels, which simplifies interconnections and reduces their fabrication costs. Although optical neural networks (ONNs) have been studied extensively, because of their complexity and bulkiness, they have not achieved commercial success comparable to that of digital electronics [1,2,7,8,9,10,11,12,13].

A previous report [14] introduced a simple and scalable optical computer, referred to as a linear combination optical engine (LCOE), to partition a neural network into linear optical and nonlinear electronic sections. Optics was used for linear calculations while electronics was employed to handle the nonlinear part, thereby maximizing the advantage of optical interconnections while maintaining parallelism. This system, based on Köhler illumination, exploits the parallelism of free-space optics and electronics such as smart pixels. A hardware demonstration involving a lens array and a spatial light modulator (SLM) showed the LCOE’s potential, and the scalability and throughput limitations were ascertained through a theoretical analysis. Furthermore, a clustering technique was proposed to overcome scalability limitations. Although the architecture’s simplicity and scalability could enhance optical parallelism, making it attractive for neural network applications with numerous synaptic connections, its use may be restricted to inference algorithms owing to the absence of a path for the backward propagation of light.

Recently, the use of a backpropagation algorithm in ONN hardware has garnered attention owing to its in situ training capability, which has the potential to increase parallel throughput and minimize errors compared with the in silico training approach of ONNs for inference [15,16]. However, since previous studies used a 4f correlator system with SLMs and coherent light sources, they inherited disadvantages stemming from this architecture [17]. In such a 4f system, the changing inputs at each neural network layer necessitate the updating of SLM pixels, which results in the use of a serial read-in mode that introduces significant delays. The switching speed of SLM pixels with current technology, such as liquid crystal display (LCD) and micro electro-mechanical systems (MEMS), is notably slower than their electronic counterparts. Moreover, this serialization would persist even if the SLM pixel-switching speed were to increase in the future. Consequently, previous systems based on 4f systems pose challenges in extending the architecture to multilayer ONNs for achieving a higher throughput. By contrast, an LCOE based on Köhler illumination does not have cascading issues and is capable of massive optical parallelism since it does not involve a coherent source or SLMs for input generation at each layer.

Here, a bidirectional optical neural network (BONN), which is a modified version of an LCOE, is proposed. A BONN is capable of providing light paths in both forward and backward directions. This bidirectional feature renders it suitable for use in algorithms such as the backpropagation algorithm, which is used for supervised learning in artificial neural networks. In the subsequent sections of this paper, BONN’s architecture is explained first, followed by a theoretical analysis of its performance and limitations.

## 2. Materials and Methods

To comprehend BONN’s architecture, one should possess a good understanding of the backpropagation algorithm [18,19], which is used in neural networks. Figure 1 presents a visual representation of this fundamental algorithm. It depicts a neural network comprising two input nodes and four output nodes interconnected by synapses, along with the mathematical representations of both types of nodes. The weighted input $z_{i}^{\left(l\right)}$ is introduced, since it will be used for theoretically explaining the backpropagation algorithm, and $\nabla \_z_{i}^{\left(l\right)}$, referred to as sensitivity, represents the gradient of the cost function with respect to the weighted input. The sensitivity of a layer is linked to the sensitivities of subsequent layers through the weights connecting them. Hence, the sensitivities of the last layer propagate backward until they reach the first layer. The computed sensitivities with respect to the weighted input are used to compute the sensitivities with respect to the weight and bias. The sensitivities are then used to adjust the current values of weights and biases in order to minimize the errors between the final output values and the target values.

Figure 2a shows the concept of the proposed BONN. The neural network depicted in Figure 1 has been transformed into a hardware schematic, incorporating components such as laser diodes (LDs), lenses, an SLM, detectors, and electronics. However, the order of the output nodes has been reversed, and the number of output nodes has been reduced to two. BONN shares similarities with the LCOE, and the main distinction lies in the inclusion of additional LDs adjacent to the photodiodes (PDs) on the second substrate, and extra PDs adjacent to the LDs on the first substrate. These supplementary LDs and PDs facilitate the creation of backward light paths that are required for implementing the backpropagation algorithm.

For the forward direction, the light paths and functions of components are similar to those of the LCOE architecture proposed in ref. [14]. An LD replaces the input node, and it emits two rays directed at lens array 1. The rays are collimated by the lens array, and the collimated rays reach the SLM. Each SLM pixel transmits a ray with a preset weight that passes through lens array 2. This lens array focuses the incident rays, with the angle of each ray depending on the position of the pixel that emitted the ray. Lens 3 sorts incoming rays into spots according to the equal-inclination rule. If the distance between the SLM and lens array 2 equals the focal length of lens 2, and the detector plane is at the focus of lens 3, the SLM and detector planes are conjugate. In this case, ray illumination areas can be clearly defined, and channel crosstalk is reduced. Detectors collect rays with equal angles from different LDs or inputs with preset weights, forming an output of a linear combination of input signals. This optical system, known as the LCOE, performs parallel, one-step calculations at the speed of light for inputs with preset weights. In the neural network community, this forward computation is termed “inference”. While the SLM-based LCOE is reconfigurable, it is better suited for inference tasks when its switching speed is lower.

For the backward direction, light is emitted by the LDs located on the second substrate, which also contains the PDs for the forward light propagation. To differentiate the LDs and PDs used for the backward direction from those used for the forward direction, the former LDs and PDs are denoted by ${LD’}_{i}^{\left(1\right)}$ and ${PD’}_{i}^{\left(1\right)}$ in Figure 2a. The rays for the backward direction are represented by thicker dashed lines (magenta and green). The LDs receive information such as $\nabla \_z_{i}^{\left(l\right)}$ from the nearby output node or, more specifically, the electronic processor. The ray emitted by the LDs passes through lens 3, lens 2, and the SLM pixel in the case of backward directions. The SLM pixels for backward direction are denoted by a primed weight parameter, and they are positioned next to those for the forward direction with the same transmission values or weight factors. It should be noted that the backward rays pass through the SLM pixels at an angle, while the forward rays are horizontal. This characteristic can be explained by considering the properties of Köhler illumination, since BONN can be considered as a modified version of LCOE, which is based on Köhler illumination [20,21]. In BONN, lenses 2 and 3 form a projection system where the SLM and the second substrate are conjugate. Since the images of SLM pixels are ${PD}_{i}^{\left(1\right)}$ and ${LD’}_{i}^{\left(1\right)}$, the distance between SLM pixels is magnified into the distance between the LDs and PDs on the second substrate. Suppose that two rays ${R’}_{00}$ and ${R’}_{00a}$ start from ${LD’}_{0}^{\left(1\right)}$ and that they cross at the pixel labeled as ${w’}_{00}^{\left(1\right)}$. An auxiliary or fictitious ray ${R’}_{00a}$ is introduced to explain the actual light path from ${LD’}_{0}^{\left(1\right)}$. Since ${R’}_{00a}$ follows a path similar to that of forward rays, it is horizontal at the SLM pixel. The angle between the two rays at the SLM plane and the second substrate is related to the magnification of the projection system. Therefore, if ${R’}_{00}$ has a certain angle from ${R’}_{00a}$ on the second substrate, it will have a certain fixed slope at the SLM plane, resulting in a fixed position of ${PD’}_{0}^{\left(0\right)}$ on the first substrate and maintaining a fixed distance from the corresponding ${LD}_{0}^{\left(0\right)}$. In this manner, backward rays from different ${LD’}_{i}^{\left(1\right)}$ can be collected by ${PD’}_{i}^{\left(0\right)}$ on the first substrate. Through the backward light paths, the sensitivity information can be transferred to the electronic processors in the previous layer. This backward information transfer renders BONN suitable for use in the backpropagation algorithm.

Since the values of $a_{j}^{\left(l-1\right)}$ and $\nabla \_z_{i}^{\left(l\right)}$ are required for the calculation of $\nabla \_w_{ij}^{\left(l\right)}$, the detectors and electronic processor are better located near $w_{ij}^{\left(l\right)}$ or ${w’}_{ij}^{\left(l\right)}$ pixels on the SLM substrate. The portion of light transmitted through these two pixels can be used to obtain the values of $a_{j}^{\left(l-1\right)}$ and $\nabla \_z_{i}^{\left(l\right)}$. The values obtained by the detectors are provided to the electronic processor for the calculation of $\nabla \_w_{ij}^{\left(l\right)}$.

More detailed features of the LD for the backward direction are shown in Figure 2b. The use of a grating and a prism facilitate the emission of multiple rays and the control of their directions. The light source can also be controlled by using diffractive optical elements (DOEs) [22], which facilitate the generation of an arbitrary number of beams in arbitrary directions. The beam divergence can be controlled by inserting a lens between the LD and the grating. The function of a lens can also be incorporated into a DOE.

The proposed BONN system can be extended to implement multilayer neural networks in the direction of beam propagation, since multiple BONN systems can be cascaded. For both forward and backward directions, the signal from the output node is directly provided to the corresponding input of the LD. Thus, two LDs for the two directions, two PDs for the two directions, and the corresponding electronic processor can form a neuron. The red dashed oval in Figure 2a shows a neuron. An example of a multilayer BONN is depicted in Figure 2c. If the system has *M* inputs, *N* outputs, and *L* layers, *M* × *N* × *L* calculations can be performed in parallel in a single step. For forward calculations, this significantly increases the throughput, which is beneficial for real-time inference applications with continuous input flow. The increase in throughput resulting from the introduction of additional layers also applies to backward calculations, provided the input is continuous.

In Figure 2d, a 3D view of a sample system with a 2 × 2 input and a 3 × 3 output is presented for the easy comprehension of the BONN setup by the reader.

Notably, incorporating incoherent light in BONN does not allow for the representation of negative weights, which are necessary for inhibitory connections in neural networks. While the use of coherent light and interference effects can enable the system to represent subtraction between inputs, such use may render the system complex and increase noise. To address this challenge, two optical channels were employed for one output, and inputs associated with positive weights were separated from those linked to negative weights, as depicted in Figure 3. This method of handling negative weights has been previously used in forward direction optical computers such as the LCOE [14] and lenslet array processors [23]. Each channel optically sums the products of input values and their weights, which is followed by electronic subtraction between the two channels. It is noteworthy that when the positive weight is used, the corresponding weight in the negative channel is set to zero, and when the negative weight is used, the corresponding weight in the positive channel is set to zero. This subtraction approach streamlines the structure at the cost of an additional channel. The use of separate channels for positive and negative weights is termed “difference mode”, while the use of a single channel (seen in Figure 2a) is termed “nondifference mode”. Since two channels are used for the forward direction and two more channels are used for backward directions, four channels or four SLM pixels are required to implement a single bidirectional channel, as shown in Figure 3.

## 3. Results

The scalability and throughput limitations of BONN were assessed using a method similar to that employed for the LCOE in Reference [14]. Since BONN shares similarities with the LCOE, its scalability limitations primarily stem from geometrical aberration rather than diffraction effects or geometric image magnification.

If a worst-case angular aberration of 1 mrad and f/# of 2 are assumed, the maximum input and output array sizes are restricted to 192 × 192. Each bidirectional channel for the difference mode comprises four subchannels or four SLM pixels, resulting in a maximum input or output array of 96 × 96 under the same optical constraints as those used in reference [14]. With these scalability considerations, the number of connections, equivalent to the number of multiply and accumulate (MAC) operations in one instruction cycle, was estimated to be approximately (96 × 96)^2^, resulting in about 8.5 × 10^7^ MAC operations. With electronic processing delays of 10 ns assumed, the optical computer’s throughput was estimated to reach 8.5 × 10^15^ MAC/s. This throughput can potentially be increased by incorporating multiple layers. While multiple layers may introduce data processing delays, they operate simultaneously, akin to pipelining in a digital computer. Since modern neural networks often involve hundreds of layers, the total throughput can be augmented accordingly. For instance, with 100 layers, the throughput can surge to 8.5 × 10^17^ MAC/s, surpassing the throughput of a tensor processing unit, which is 4.59 × 10^14^ floating point operations/s [24].

However, the above parallel throughput estimation is for the assumption that the input flow is continuous, which is typical for forward direction data flow in inference applications. For the backpropagation algorithm, the data flow is not continuous but occurs in a single pass, which is sequential. Therefore, the parallel throughput does not increase with the number of layers. In practice, the maximum throughput for backward data flow is only 8.5 × 10^15^ MAC/s. The performance of various architectures of neural networks is compared in Table 1. Evidently, BONN has greater potential than other ONNs and electronic competitors.

SLMs consume insignificant power, since the typical liquid crystal consumes around 10 W/m^2^ [25]. Therefore, most of the power consumed by BONN may be used by the light source for generating inputs and by the electronic processors. If vertical-cavity surface-emitting lasers with an operating power of 1 mW are used for light sources for generating 192 × 192 inputs, one layer of BONN consumes about 70 W [26]. This amount of heat can be dissipated through natural convection or forced convection without a significant increase in the device temperature, unless the size of the board or the substrate, including light sources, is too small.

There is another major problem in realizing BONN in hardware. Training the neural network requires that the data flow forward and backward in turn numerous times. Consequently, the pixel values of the SLM should be updated whenever the direction of data flow changes. The time span between the updating processes is approximately the product of the delay time of the electronics in a single layer and the number of layers. For instance, if the system comprises 100 layers, the refresh time would be about 1 μs.

Nevertheless, the majority of currently available SLMs exhibit sluggish performance, leading to significant latency [27,28], and the low refresh rate of SLMs significantly affects the throughput of BONN. Despite advances in MEMS technology, most SLMs still operate within a range of tens of kilohertz, significantly slower than electronic switches, which introduces substantial latency in BONN operations.

An alternative approach to realizing BONN involves the development of high-speed SLMs based on absorption modulation [29]. However, developing a novel SLM device is extremely challenging, requiring considerable time and effort. Since this BONN architecture demonstrates the feasibility of a high-throughput ONN in theory, it may serve as an example of applications that could require the future development of high-speed SLMs.

**Table 1 micromachines-15-00701-t001:** Comparison of the performance of various neural network architectures.

Architecture	Computing Speed (TOPs/s)	Features
Diffractive ONN [15]	7.86 × 10^−4^	Backpropagation algorithm 4f correlator 1 mW laser source
Tensor Processing Unit [24]	4.59 × 10^2^	Electronic AI accelerator TPUv5p
Photonic Reservoir Computing [30]	100	Scattering medium
BONN	8.5 × 10^3^	Bidirectional data flow; for inference, the speed may be increased further

## 4. Discussion

Besides its use in implementing the backpropagation algorithm, BONN can be used to implement a compact ONN for inference applications. The simplest example of a compact ONN can be realized using a two-layer or two-mirror-like BONN (TMLBONN), where the data flow back and forth between two layers, emulating *L_T_* layer LCOE, updating the weights read from the memory inside electronic processors connected to the SLM pixels. Although the parallel throughput of the TMLBONN for inference is reduced by a factor of *L_T_* compared with the *L_T_* layer LCOE that works simultaneously for continuous input, the volume of the ONN hardware is also reduced by a factor of *L_T_*.

Since the TMLBONN is cascadable for an odd *L_T_*, it can be extended into multiple TMLBONN structures to increase the parallel throughput for continuous input data. For example, if a TMLBONN can emulate a five-layer BONN by bouncing data back and forth between two layers, 20 cascaded TMLBONNs can replace a 100-layer BONN. Since the input and output layers of a cascaded TMLBONN can be installed on a single substrate, the actual number of layers in 20 cascaded TMLBONNs is 20, excluding the zeroth layer. The use of the TMLBONN reduces the volume of the system by about five times compared with a 100-layer BONN. In terms of parallel throughput, 20 cascaded TMLBONNs show higher performance than a single TMLBONN by a factor of 20, as all 20 TMLBONNs operate simultaneously. Since the time delay of a TMLBONN increases five times, its performance gain is four times that of a single-layer BONN. Therefore, if a BONN has a throughput of 8.5 × 10^15^ MAC/s, 20 cascaded TMLBONNs with five foldings have a throughput of 3.4 × 10^16^ MAC/s and hardware that is five times more compact than 100-layer LCOEs. This compact structure is recommended for form-factor critical applications and in the early stage of the construction of BONN, since a two-layer system is cheaper and easier to assemble than a multi-layer system.

In a previous report [14], the use of the clustering technique was suggested to overcome the scaling limit of LCOE. It is worth investigating the possibility of using the clustering technique in the BONN architecture, since the technique may provide scalability and a corresponding increase in the parallel throughput as the scaling proceeds, at least theoretically.

The clustering technique creates a new optical module with doubled input and output array sizes by assembling basic optical modules. Similar to mathematical induction, clustering can continue and increase the number of connections and throughput by four times at each step. The simplest basic optical module has two input and two output ports, similar to the optical module shown in Figure 2a. Since BONN is bidirectional, an output port can be used as an input port with reverse data flow. If reverse data flow occurs in a clustering structure as observed in reference [14], data flow starts from the new input (old output) ports of the clustering structure, passes through splitting, cross, and straight-through connections, and full interconnection modules. The final data appear at the new output (old input) ports with the weighted sum of all input ports, resulting in full connection between the doubled input and output ports under reverse data flow. Thus, the clustering technique is applicable to BONN and can be used to overcome the scaling limit of BONN.

## 5. Conclusions

BONN is proposed for providing backward connections in addition to the forward connections available in the LCOE architecture. The use of the properties of Köhler illumination in BONN facilitates the addition of LDs and PDs to provide optical channels for backward direction. BONN’s bidirectional nature can support algorithms such as the backpropagation algorithm in artificial neural networks.

The scaling limit of BONN is about (96 × 96) for input and output arrays under the same optical constraints used for the analysis of the LCOE in reference [14]. If electronic processing delays of 10 ns are assumed, the throughput of BONN can reach 8.5 × 10^15^ MAC/s. While the BONN throughput can increase with the number of layers for continuous input, since the backpropagation algorithm’s data flow occurs in a single pass, the increase in the parallel throughput with the number of layers is limited. The maximum throughput for backward data flow is capped at 8.5 × 10^15^ MAC/s.

The successful implementation of the backpropagation algorithm by using BONN depends on the development of fast SLM devices, since neural network training requires frequent changes in data flow between forward and backward directions. Such frequent changes require the SLM pixel value to be updated. A 100-layer system requires a refresh time on the order of microseconds, while currently available SLMs with LCD and MEMS technology have longer switching times. Developing SLMs based on absorption modulation could be a solution, but it would require significant time and effort.

Another application of BONN that was considered was in inference to realize small form factor hardware. TMLBONN can emulate a multilayer BONN through data transmission back and forth between two layers, albeit with a longer time delay. More bounces or foldings between the two layers can save hardware space by the same factor as the number of foldings. Multiple TMLBONNs can be cascaded to increase the parallel throughput. The use of multiple TMLBONN structures can reduce the hardware volume and provide reasonable performance gain.

Finally, the possibility of applying the clustering technique to BONN to overcome the scaling limit was investigated. BONN’s bidirectionality caused the basic optical module of the cluster to operate backward. The reverse data flow provided full interconnections for the backward direction between the doubled new input and output ports. Owing to its bidirectionality, BONN is expected to be widely used in algorithms such as the backpropagation algorithm, employed for supervised learning in artificial neural networks. Furthermore, the TMLBONN structure can help increase the compactness of inference hardware.

## Figures and Tables

**Figure 1 micromachines-15-00701-f001:**
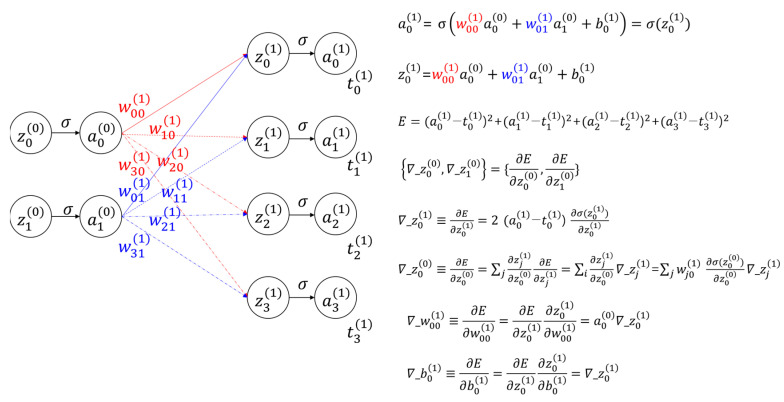
Example of a simple backpropagation algorithm used in a neural network, along with mathematical formulae; $a_{i}^{\left(l\right)}$ is the *i*th input or output node in the *l*th layer, $w_{ij}$ is the weight connecting the *j*th input node and the *i*th output node, $b_{i}$ is the *i*th bias, $z_{i}^{\left(l\right)}$ is the weighted input to the neurons in the *l*th layer, *E* is the cost function, $t_{i}^{\left(l\right)}$ is the target value; ∇_ (referred to as sensitivity) represents the gradient of the cost function with respect to the following variable, and σ is a sigmoid function.

**Figure 2 micromachines-15-00701-f002:**
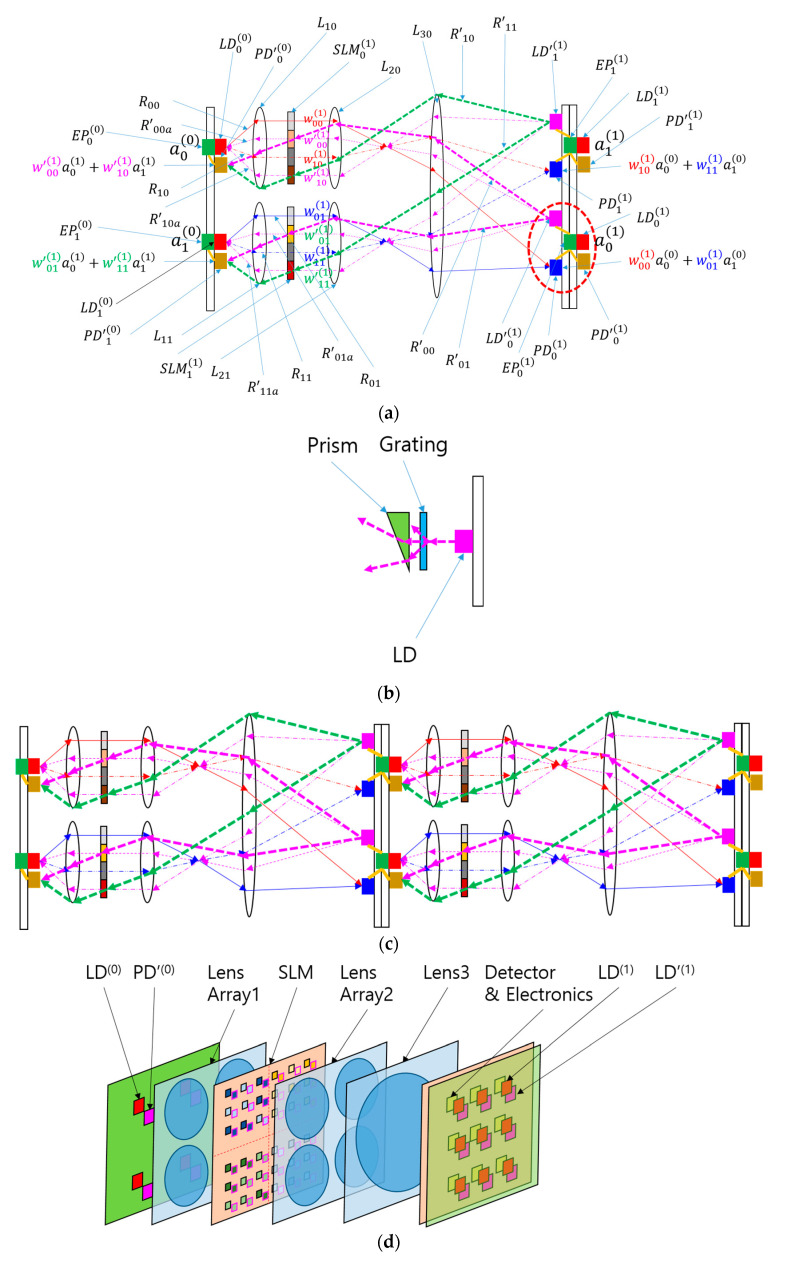
Bidirectional optical neural network (BONN) based on Köhler illumination and free-space optics. It consists of lens arrays and a spatial light modulator. (**a**) A schematic of BONN, and the related mathematical formulae. ${LD}_{i}^{\left(l\right)}$, ${PD}_{i}^{\left(l\right)}$, and $R_{jk}$ denote the *i*th laser diode, photodiode, and ray in the *l*th layer for the forward direction, respectively, and ${LD’}_{i}^{\left(l\right)}$, ${PD’}_{i}^{\left(l\right)}$, and ${R’}_{jk}$ represent the *i*th laser diode, photodiode, and ray in the *l*th layer for the backward direction, respectively. $L_{1p}$, $L_{2p}$, and $L_{30}$ represent lens array 1, lens array 2, and lens 3, respectively. ${EP}_{i}^{\left(l\right)}$ and ${SLM}_{i}^{\left(l\right)}$ represent the *i*th electronic processor and the spatial light modulator (SLM) in the *l*th layer, respectively. The red dashed oval indicates a neuron. (**b**) A schematic of the laser diode (${LD’}_{i}^{\left(1\right)})$ for the backward direction; the abbreviation LD represents “laser diode”. (**c**) A cascadable multilayer BONN. (**d**) Three-dimensional view of a BONN system with a 2 × 2 input and a 3 × 3 output.

**Figure 3 micromachines-15-00701-f003:**
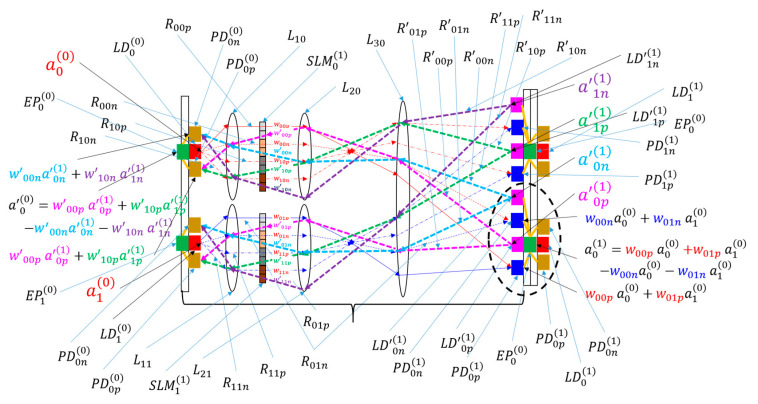
Schematic of difference mode BONN. Symbols such as $X_{yp}$ with a subscript ending with “*p*” pertain to the positive channel, while those such as $X_{yn}$ with a subscript ending with “*n*” represent the negative channel. Otherwise, the notation is identical to that used in Figure 2a. The black dashed oval shows a neuron.

## Data Availability

The datasets generated during and/or analyzed during the current study are available from the corresponding author on reasonable request.

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
