# Peer review of "Bidirectional Optical Neural Networks Based on Free-Space Optics Using Lens Arrays and Spatial Light Modulator"

_micromachines, 2024, doi:10.3390/mi15060701_

Round 1

Reviewer 1 Report

Comments and Suggestions for Authors

The manuscript proposes a novel architecture for a bidirectional optical neural network (BONN), leveraging free-space optics and components such as lens arrays and spatial light modulators to enable backward data flow for the backpropagation algorithm in neural networks. However, some points remain unclear to me. I would appreciate further clarification from the author before recommending publication in Micromachines.

1. I think there is a question that readers are most likely concerned with: compared to current state-of-the-art electronic neural network implementations, what are the energy efficiency and speed of the BONN?

2. The BONN system involves optical systems. However, we know that optical systems are often susceptible to noise and signal degradation. How does BONN handle errors or noise in the signal, especially given the complexity of maintaining bidirectional data flow?

3. Can the author provide any experimental results to support the learning capabilities of BONN?

4. In the manuscript, the author mentions Köhler illumination. Could the author expand on how this impacts the design and functionality of BONN compared to other optical systems that do not utilize this method?

5. I know that dual-layer spatial light modulators (SLM) can modulate incoming polarized light. Has the author considered integrating the polarization degree of freedom into the BONN architecture? It may improve the computing speed or throughput.

Author Response

Comments 1: The manuscript proposes a novel architecture for a bidirectional optical neural network (BONN), leveraging free-space optics and components such as lens arrays and spatial light modulators to enable backward data flow for the backpropagation algorithm in neural networks. However, some points remain unclear to me. I would appreciate further clarification from the author before recommending publication in Micromachines.

  I think there is a question that readers are most likely concerned with: compared to current state-of-the-art electronic neural network implementations, what are the energy efficiency and speed of the BONN?

Response 1: Thank you for pointing this out. We agree with this comment. Therefore, the following sentences about energy consumption were added, along with a table comparing the performance of various architectures, including electronic neural networks.

the revised version (p7, line251)

“The performance of various architectures of neural networks is compared in Table 1. Evidently, BONN has greater potential than other ONNs and electronic competitors.

SLMs consume insignificant power since the typical liquid crystal consumes around 10 W/m2 [25]. Therefore, most of the power consumed by BONN may be used by the light source for generating inputs and by the electronic processors. If VCSELs with an operating power of 1 mW are used for light sources for generating  192 × 192 inputs, one layer of BONN consumes about 70 W [26]. This amount of heat can be dissipated through natural convection or forced convection without a significant increase in the device temperature, unless the size of the board or the substrate, including light sources, is too small.”

Table 1. Comparison of the performance of various neural network architectures.

Architecture

Computing speed (TOPs/s)

Features

Diffractive ONN [15]

Tensor Processing Unit [24]

Photonic Reservoir Computing [30]

7.86×10-4

4.59×102

100

Backpropagation algorithm

4f correlator

1 mW laser source

Electronic AI accelerator

TPUv5p

Scattering medium

BONN

8.5×103

Bidirectional data flow;

for inference, the speed may be increased further.

Comments 2: The BONN system involves optical systems. However, we know that optical systems are often susceptible to noise and signal degradation. How does BONN handle errors or noise in the signal, especially given the complexity of maintaining bidirectional data flow?

Response 2: In general, the optical neural network is known to have less electromagnetic crosstalk noise than electronic connections. This is one of the reasons why many researchers have been trying to use optical interconnection instead of electronic connections. We explain this idea in more detail in the introduction. Furthermore, the OCNN and BONN, based on Köhler illumination, have less optical crosstalk due to their clear imaging capability of SLM pixels into the detector array. This point is explained in more detail in the previous report [Ref. 14]. We display the part of the manuscript regarding the optical crosstalk of BONN in the following.

the revised version (p1, line34)

“Free-space optics [3] and smart pixels [5] are promising options for overcoming the limitations of electronic processors, with the former being especially beneficial for artificial neural networks [6]. Purely electrical neural networks have a  complex topology of connections as the number of input and output nodes increases. Moreover, the complex topology and proximity between electrical wires may cause significant electromagnetic crosstalk or noise, especially at high clock rates [3, 4]. Unlike electrical interconnections, light paths in free-space optics can cross freely without electromagnetic crosstalk noise between channels, which simplifies interconnections and reduces their fabrication costs.”

(Regarding the optical crosstalk of BONN) (p3, line117)

“If the distance between the SLM and lens array 2 equals the focal length of lens 2 and the detector plane is at the focus of lens 3, the SLM and detector planes are conjugate. In this case, ray illumination areas can be clearly defined and channel crosstalk is reduced.”

(Regarding the optical crosstalk of BONN) (p4, line138)

“This characteristic can be explained by considering the properties of Köhler illumination, since BONN can be considered as a modified version of LCOE, which is based on Köhler illumination [20, 21]. In BONN, lenses 2 and 3 form a projection system where the SLM and the second substrate are conjugate.”

Comments 3: Can the author provide any experimental results to support the learning capabilities of BONN?

Response 3: Unfortunately, we could not perform experiments with this architecture.

the revised version (p2, line78)

“In the subsequent sections of this paper, BONN's architecture is explained first, followed by a theoretical analysis of its performance and limitations.”

Comments 4: In the manuscript, the author mentions Köhler illumination. Could the author expand on how this impacts the design and functionality of BONN compared to other optical systems that do not utilize this method?

Response 4: According to the reviewer’s comment, we added a paragraph comparing BONN with the other optical system.

the revised version (p2, line58)

“Recently, the use of a backpropagation algorithm in ONN hardware has garnered attention owing to its in situ training capability, which has the potential to increase parallel throughput and minimize errors compared with the in silico training approach of ONNs for inference [15, 16]. However, since previous studies used a 4f correlator system with SLMs and coherent light sources, they inherited disadvantages stemming from this architecture [17]. In such a 4f system, the changing inputs at each neural network layer necessitate the updating of SLM pixels, which results in the use of a serial read-in mode that introduces significant delays. The switching speed of SLM pixels with current technology, such as liquid crystal display (LCD) and micro electro-mechanical systems (MEMS), is notably slower than their electronic counterparts. Moreover, this serialization would persist even if the SLM pixel switching speed were to increase in the future. Consequently, previous systems based on 4f systems pose challenges in extending the architecture to multilayer ONNs for achieving a higher throughput. By contrast, an LCOE based on Köhler illumination does not have cascading issues and is capable of massive optical parallelism since it does not involve a coherent source or SLMs for input generation at each layer.”

(Regarding the Köhler illumination of BONN) (p3, line117)

“If the distance between the SLM and lens array 2 equals the focal length of lens 2 and the detector plane is at the focus of lens 3, the SLM and detector planes are conjugate. In this case, ray illumination areas can be clearly defined and channel crosstalk is reduced.”

(Regarding the Köhler illumination of BONN) (p4, line138)

“This characteristic can be explained by considering the properties of Köhler illumination, since BONN can be considered as a modified version of LCOE, which is based on Köhler illumination [20, 21]. In BONN, lenses 2 and 3 form a projection system where the SLM and the second substrate are conjugate.”

Comments 5: I know that dual-layer spatial light modulators (SLM) can modulate incoming polarized light. Has the author considered integrating the polarization degree of freedom into the BONN architecture? It may improve the computing speed or throughput.

Response 5: We greatly appreciate the reviewer’s suggestion. However, we are not familiar with dual-layer spatial light modulators and do not have good idea of how to incorporate this idea into BONN. We will consider this in the next study or paper. 

Reviewer 2 Report

Comments and Suggestions for Authors

I have the following remarks:

1) I am not sure why author is comparing the current work with his own work (ref 11) and providing an advancement to the solution. Why there is no comparison with works from other authors who conducted the work on similar topic? 

2) Is there an experimental demonstration of this work? Or it is just analytical/numerical work?

3) The literature review is scarce and outdated. The author should add recent works published in current year to show the progress of the topic. For instance: https://www.mdpi.com/2079-4991/14/8/697; https://pubs.acs.org/doi/10.1021/acsphotonics.2c01516.

4) In the paper especially in the Conclusion section, the author has repeatedly used the word "We", whereas in the paper, there is only a single author. Therefore, I suggest formulating the sentences for better understanding. 

5) A comparison table should be added at the end of the paper to compare the ONN proposed by other researchers and the work presented in this work. Also, highlight the novelty and accuracy of this work. 

6) The introduction section has to be written in detail with an additional literature review. Currently, it has only one paragraph, after that author starts talking about his previous paper. 

7) The potential applications of BONNs should be explain in detail.

Comments on the Quality of English Language

none. 

Author Response

Comments 1:  I have the following remarks:

1) I am not sure why author is comparing the current work with his own work (ref 11) and providing an advancement to the solution. Why there is no comparison with works from other authors who conducted the work on similar topic?

Response 1: Thank you for pointing this out. I agree with this comment. Therefore, I have added the works from other authors who studied training using optical neural networks.

the revised version (p2, line58)

“Recently, the use of a backpropagation algorithm in ONN hardware has garnered attention owing to its in situ training capability, which has the potential to increase parallel throughput and minimize errors compared with the in silico training approach of ONNs for inference [15, 16]. However, since previous studies used a 4f correlator system with SLMs and coherent light sources, they inherited disadvantages stemming from this architecture [17]. In such a 4f system, the changing inputs at each neural network layer necessitate the updating of SLM pixels, which results in the use of a serial read-in mode that introduces significant delays. The switching speed of SLM pixels with current technology, such as liquid crystal display (LCD) and micro electro-mechanical systems (MEMS), is notably slower than their electronic counterparts. Moreover, this serialization would persist even if the SLM pixel switching speed were to increase in the future. Consequently, previous systems based on 4f systems pose challenges in extending the architecture to multilayer ONNs for achieving a higher throughput. By contrast, an LCOE based on Köhler illumination does not have cascading issues and is capable of massive optical parallelism since it does not involve a coherent source or SLMs for input generation at each layer.”

[15] Zhou, T.; Fang, L.; Yan, T.; Wu, J.; Li, Y.; Fan, J.; Wu, H.; Lin, X.; Dai, Q. In situ optical backpropagation training of diffractive optical neural networks. Photonics Research, 2020, 8(6), pp.940-953.

[16] Gu, Z.; Huang, Z.; Gao, Y.; Liu, X. Training optronic convolutional neural networks on an optical system through backpropagation algorithms. Optics Express, 2022, 30(11), 19416-19440.

Comments 2: Is there an experimental demonstration of this work? Or it is just analytical/numerical work?

Response 2: Unfortunately, we could not perform experiments with this architecture.

the revised version (p2, line78)

“In the subsequent sections of this paper, BONN's architecture is explained first, followed by a theoretical analysis of its performance and limitations.”

Comments 3: The literature review is scarce and outdated. The author should add recent works published in current year to show the progress of the topic. For instance: https://www.mdpi.com/2079-4991/14/8/697; https://pubs.acs.org/doi/10.1021/acsphotonics.2c01516.

Response 3: According to the reviewer’s comment, we have included the above two references and more.

the revised version (p11, line375)

[7] Khonina, S.N.; Kazanskiy, N.L.; Skidanov, R.V.; Butt, M.A. Exploring Types of Photonic Neural Networks for Imaging and Computing—A Review Nanomaterials, 2024, 14, 697. https://doi.org/10.3390/nano14080697

[8] Liao, K.; Dai, T.; Yan, Q.; Hu, X.; Gong, Q. Integrated Photonic Neural Networks: Opportunities and Challenges. ACS Photonics 2023 10 (7), 2001-2010 DOI: 10.1021/acsphotonics.2c01516

[13] Lin, X.; Rivenson, Y.; Yardimci, N. T.; Veli, M.; Luo, Y.; Jarrahi, M.; Ozcan, A. All-optical machine learning using diffractive deep neural networks. Science, 2018, 361(6406), 1004-1008.

Comments 4: In the paper especially in the Conclusion section, the author has repeatedly used the word "We", whereas in the paper, there is only a single author. Therefore, I suggest formulating the sentences for better understanding.

Response 4: According to the reviewer’s comment, I have replaced all sentences including 'we' with those without it.

(the previous version)

“We propose BONN to provide backward connections in addition to the forward con-292 nections available in the LCOE architecture.”

-> the revised version (p9, line324)

“BONN is proposed for providing backward connections in addition to the forward connections available in the LCOE architecture.”

(the previous version)

“We considered another application of BONN: in inference to realize small form factor 311 hardware.”

-> the revised version (p9, line343)

“Another application of BONN that was considered was in inference to realize small form factor hardware.”

(the previous version)

“Finally, we investigated the possibility of applying the clustering technique to BONN 318 to overcome the scaling limit.”

-> the revised version (p10, line350)

“Finally, the possibility of applying the clustering technique to BONN to overcome the scaling limit was investigated.”

(the previous version)

“We expect that owing to its bidirectionality, BONN will be widely used in algorithms such as the back-propagation algorithm, which is used for supervised learning in artificial neural networks.”

-> the revised version (p10, line353)

“Owing to its bidirectionality, BONN is expected to be widely used in algorithms such as the backpropagation algorithm, employed for supervised learning in artificial neural networks.”

Comments 5: A comparison table should be added at the end of the paper to compare the ONN proposed by other researchers and the work presented in this work. Also, highlight the novelty and accuracy of this work.

Response 5: I added a table comparing the performance of other architectures and highlighted the novelty of this work.  

the revised version (p7, line251)

“The performance of various architectures of neural networks is compared in Table 1. Evidently, BONN has greater potential than other ONNs and electronic competitors.”

Table 1. Comparison of the performance of various neural network architectures.

Architecture

Computing speed (TOPs/s)

Features

Diffractive ONN [15]

Tensor Processing Unit [24]

Photonic Reservoir Computing [30]

7.86×10-4

4.59×102

100

Backpropagation algorithm

4f correlator

1 mW laser source

Electronic AI accelerator

TPUv5p

Scattering medium

BONN

8.5×103

Bidirectional data flow;

for inference, the speed may be increased further.

Comments 6: The introduction section has to be written in detail with an additional literature review. Currently, it has only one paragraph, after that author starts talking about his previous paper.

Response 6: According to the reviewer’s comment, I have corrected the introduction with more literature review.  

the revised version (p1, line36)

Purely electrical neural networks have a  complex topology of connections as the number of input and output nodes increases. Moreover, the complex topology and proximity between electrical wires may cause significant electromagnetic crosstalk or noise, especially at high clock rates [3, 4]. Unlike electrical interconnections, light paths in free-space optics can cross freely without electromagnetic crosstalk noise between channels, which simplifies interconnections and reduces their fabrication costs.

the revised version (p1, line58)

Recently, the use of a backpropagation algorithm in ONN hardware has garnered attention owing to its in situ training capability, which has the potential to increase parallel throughput and minimize errors compared with the in silico training approach of ONNs for inference [15, 16]. However, since previous studies used a 4f correlator system with SLMs and coherent light sources, they inherited disadvantages stemming from this architecture [17]. In such a 4f system, the changing inputs at each neural network layer necessitate the updating of SLM pixels, which results in the use of a serial read-in mode that introduces significant delays. The switching speed of SLM pixels with current technology, such as liquid crystal display (LCD) and micro electro-mechanical systems (MEMS), is notably slower than their electronic counterparts. Moreover, this serialization would persist even if the SLM pixel switching speed were to increase in the future. Consequently, previous systems based on 4f systems pose challenges in extending the architecture to multilayer ONNs for achieving a higher throughput. By contrast, an LCOE based on Köhler illumination does not have cascading issues and is capable of massive optical parallelism since it does not involve a coherent source or SLMs for input generation at each layer.

[15] Zhou, T.; Fang, L.; Yan, T.; Wu, J.; Li, Y.; Fan, J.; Wu, H.; Lin, X.; Dai, Q. In situ optical backpropagation training of diffractive optical neural networks. Photonics Research, 2020, 8(6), pp.940-953.

[16] Gu, Z.; Huang, Z.; Gao, Y.; Liu, X. Training optronic convolutional neural networks on an optical system through backpropagation algorithms. Optics Express, 2022, 30(11), 19416-19440.

[17] Ju, Y.G., Scalable Optical Convolutional Neural Networks Based on Free-Space Optics Using Lens Arrays and a Spatial Light Modulator. Journal of Imaging, 2023, 9(11), p.241.

Comments 7: The potential applications of BONNs should be explain in detail.

Response 7: We have included the potential applications of BONNs as follows.  

the revised version (p2, line58)

“Recently, the use of a backpropagation algorithm in ONN hardware has garnered attention owing to its in situ training capability, which has the potential to increase parallel throughput and minimize errors compared with the in silico training approach of ONNs for inference [15, 16].”

(p9, line324)

“BONN is proposed for providing backward connections in addition to the forward connections available in the LCOE architecture. The use of the properties of Köhler illumination in BONN facilitates the addition of LDs and PDs to provide optical channels for backward direction . BONN’s bidirectional nature can support algorithms such as the backpropagation algorithm in artificial neural networks.”

(p9, line343)

“Another application of BONN that was considered was in inference to realize small form factor hardware. TMLBONN can emulate a multilayer BONN through data transmission back and forth between two layers, albeit with a longer time delay. More bounces or foldings between the two layers can save hardware space by the same factor as the number of foldings. Multiple TMLBONNs can be cascaded to increase the parallel throughput. The use of multiple TMLBONN structures can reduce the hardware volume and provide reasonable performance gain.

Finally, the possibility of applying the clustering technique to BONN to overcome the scaling limit was investigated. BONN’s bidirectionality caused the basic optical module of the cluster to operate backward. The reverse data flow provided full interconnections for the backward direction between the doubled new input and output ports. Owing to its bidirectionality, BONN is expected to be widely used in algorithms such as the backpropagation algorithm, employed for supervised learning in artificial neural networks. Furthermore, the TMLBONN structure can help increase the compactness of inference hardware.”

4. Response to Comments on the Quality of English Language

Point 1: Moderate editing of English language required.

Response 1: The manuscript has been professionally translated into English, and the certificate will be included.

Round 2

Reviewer 1 Report

Comments and Suggestions for Authors

The author addressed all my concerns. Thank you for the author's effort.

Reviewer 2 Report

Comments and Suggestions for Authors

I am willing to accept the paper in its current form.